# Wetting ridge assisted programmed magnetic actuation of droplets on ferrofluid-infused surface

Jianqiang Zhang [1], Xuejiao Wang[1], Zhaoyue Wang[1], Shangfa Pan[2], Bo Yi [1], Liqing Ai[1], Jun Gao [2✉], Frieder Mugele [3✉] & Xi Yao [1,4✉]

Flexible actuation of droplets is crucial for biomedical and industrial applications. Hence, various approaches using optical, electrical, and magnetic forces have been exploited to actuate droplets. For broad applicability, an ideal approach should be programmable and be able to actuate droplets of arbitrary size and composition. Here we present an "additive-free" magnetic actuation method to programmably manipulate droplets of water, organic, and biological fluids of arbitrary composition, as well as solid samples, on a ferrofluid-infused porous surface. We specifically exploit the spontaneously formed ferrofluid wetting ridges to actuate droplets using spatially varying magnetic fields. We demonstrate programmed processing and analysis of biological samples in individual drops as well as the collective actuation of large ensembles of micrometer-sized droplets. Such model respiratory droplets can be accumulated for improved quantitative and sensitive bioanalysis - an otherwise prohibitively difficult task that may be useful in tracking coronavirus.

[1] Department of Biomedical Sciences, City University of Hong Kong, Hong Kong, P. R. China. [2] Qingdao Institute of Bioenergy and Bioprocess Technology, Chinese Academy of Sciences, 266101 Qingdao, P. R. China. [3] Physics of Complex Fluids, MESA+ Institute for Nanotechnology, University of Twente, P.O. Box 217, 7500AE Enschede, The Netherlands. [4] Shenzhen Research Institute, City University of Hong Kong, 518075 Shenzhen, P. R. China. ✉email: jun.gao@qibebt.ac.cn; f.mugele@utwente.nl; xi.yao@cityu.edu.hk

n a broad range of applications including microfluidics[1,2], bioanalysis[3], optofluidics[4], and heat transfer[5–7], flexible and programable actuation of droplets is highly demanded. To this end, a wide range of droplet actuation methods using optical[8–10], electrical[11], and magnetic[12–18] as well as chemical cues[6] have been developed. Successful approaches should allow for strong, reversible, and freely programmable droplet actuation while imposing as little restrictions as possible on droplet composition and sizes. For example, for microfluidics and bioanalysis, the approach should be compatible to biological fluids without suffering from fouling. For high precision analysis, the actuation of micro-sized droplets is needed, while for other applications the actuation of large droplets is required. However, the above requirements have remained challenging to fulfill. For example, electrowetting-on-dielectric, the most successful electrical droplet actuation method[11,19,20], still suffers from long-term stability problems caused by electrical degradation[21–23] and biofouling[24] on intrinsically hydrophobic substrates. Alternatively, electro-dewetting on hydrophilic substrates was proposed[25]. The direct contact between electrode and droplet, however, may cause electrochemical reactions. Contactless light[8–10] and magnetic[12–18] actuation methods may circumvent these problems. Yet, most light-based methods[9,10] suffer from slow droplet transport, due to the low actuation force provided by light. On the contrary, magnetic actuation, widely used in microfluidics and bioanalysis[12], provides large force, and thus potentially fast droplet transport. Moreover, the electromagnets can be miniaturized down to microscale[26,27], holding promise for digital magnetic microfluidics[28,29]. However, magnetic actuation generally necessitates magnetic additives in the droplets[28,30], which causes sample contamination concern. To address this problem, one could instead make the substrate magnetically active[13–18]. So far, however, the control offered by these methods has been insufficient for digital microfluidics and other automated droplet operations.

In this work, we report a method for wetting ridge assisted programed (WRAP) magnetic actuation of liquid droplets on liquid-infused surfaces that is easily reconfigurable and displays anti-fouling properties. Contrary to most previous actuation methods that directly drive droplets, our WRAP method manipulates the ferrofluid wetting ridge which "wraps" and carries the droplet. Thanks to this indirect actuation, the method is tolerant to variable droplet compositions and properties. It allows to actuate a wide range of sizes of droplets of water, organic fluids, as well as complex biological materials as long as they are immiscible with the infusion liquid. In particular, WRAP does not require magnetic additives in the sample drop.

## Results and discussions

**Actuation of droplets with WRAP.** To demonstrate the basic functionality of WRAP, we sprayed water droplets with diameters from ~10 μm to several hundreds of μm (Supplementary Fig. 1) onto a liquid-infused porous surface with ~2° contact angle hysteresis. The porous substrate (Supplementary Fig. 2) was infused with a ferrofluid consisting of silicone oil with $Fe_3O_4$ nanoparticles (see Methods). Characterization of the ferrofluid properties is shown in Supplementary Fig. 3. An excess of several tens of micrometers of this fluid was kept on the surface. Together with the positive spreading parameter and the resulting cloaking layer of ferrofluid on the water[31,32] (see Supplementary Table 1), this excess ensures that saturated wetting ridges form independently around each drop[33]. The infused ferrofluid imparts the surface with magnetic activity[15–18]. Upon turning on an electromagnet (see Methods for details) underneath the sample, all droplets within a collection area of several cm² were quickly set in motion and accumulated within a single big drop on top of the magnet (Fig. 1a, Supplementary Video 1). By contrast, without magnetic actuation, the droplets could only randomly coalesce over the whole surface (Supplementary Fig. 4).

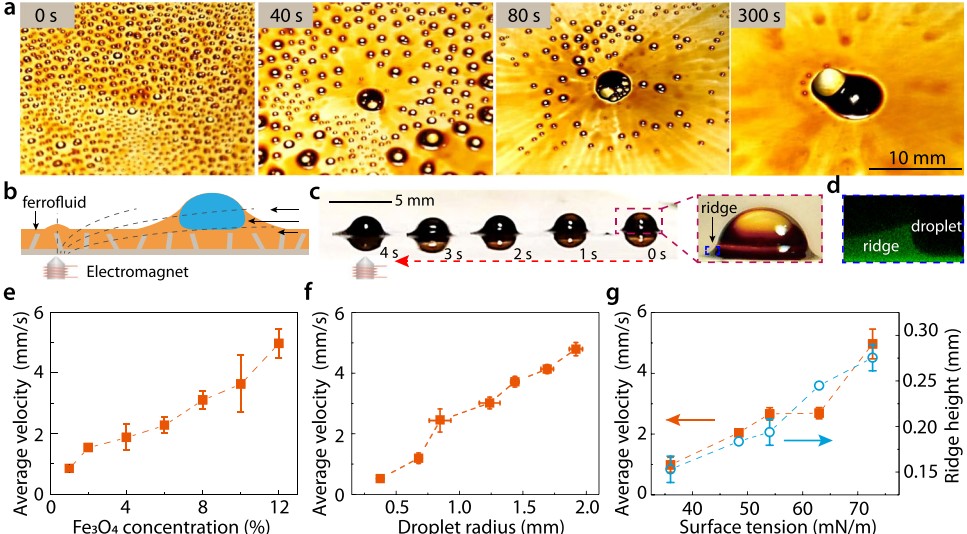

**Fig. 1 Magnetic actuation of additive-free droplets on ferrofluid-infused slippery surface. a** Collective actuation of numerous micro-sized droplets. The growing black part at the center was the ferrofluid (12 v/v% $Fe_3O_4$ nanoparticles) raised by the electromagnet. **b** The actuation force is provided by the ferromagnetic wetting ridge that spontaneously forms around the deposited droplet. The dotted lines and the solid arrows illustrate the magnetic induction lines and the droplet moving direction, respectively. **c** Snapshots of a typical actuation process. Red dotted arrow indicates the droplet moving direction. Inset: magnified optical image of the deposited droplet (rose red dashed rectangle), showing the wetting ridge (blue dashed rectangle). **d** Confocal fluorescence image of the wetting ridge. **e** The driving velocity increases with the concentration of the magnetic particles ($Fe_3O_4$ nanoparticles) in the ferrofluid (drop volume: 2 μL). **f** The velocity also increases with the droplet radius for different liquids, presumably because the ridge volume increases with the droplet size, and **g** it increases with the surface tension of the droplet (orange solid squares), presumably because a liquid with higher surface tension generally raises higher ridge (blue open squares). Dashed lines are eye-guide. The error bars represent the standard deviation ($n = 3$).

Therefore, WRAP has the capability to collect large amounts of droplets with a random initial position and broad size distribution using a single actuator. As we will show below, this capability enables enhanced sensitivity in the biomedical detection of airborne pathogens.

Before exploring these aspects, we focus on the actuation mechanism of WRAP. Optical imaging and high-resolution confocal fluorescence microscopy imaging both confirmed the spontaneous formation of a ferrofluid wetting ridge around each droplet (Fig. 1b–d). This wetting ridge is pulled up by capillary forces[34]. Its volume increases with increasing drop size. Far away from the magnet, its shape is governed by the balance of capillarity and gravity. However, being made out of ferrofluid, the wetting ridge is also subject to magnetic forces. This becomes particularly apparent directly on top of the magnet, where the magnetic stress competes with both capillarity and gravity to form a characteristic protrusion[35] (Fig. 1b, left, Supplementary Fig. 5). Note that surface instability was not observed, which is explained in Supplementary Note 1). Far away from the magnet, however, magnetic forces are weak and the meniscus merely experiences a net force pulling it against the direction of the magnetic field gradient, as any dipole in an inhomogeneous external field. Such magnetic forces, drive the droplet (Fig. 1c, Supplementary Video 2). The ferrofluid infused in the porous surface also experiences magnetic forces, but its movement is significantly slower, and thus plays a minor role in the droplet actuation process (Supplementary Fig. 6). To explore the actuation force in more detail, we deposited droplets of variable size and surface tension on the surface and monitored their average velocity upon activating the magnet for ferrofluids with a variable weight concentration of $Fe_3O_4$ nanoparticles. Upon activating the electromagnet, the droplets were immediately set in motion towards the magnet. The velocity, averaged over a travel distance of ~2 cm, was found to increase with the concentration of $Fe_3O_4$ particles in the ferrofluid (Fig. 1e), suggesting that the magnetic force scales as the total magnetization of the wetting ridge. As we increase the drop size, we found the average speed to increase with the drop radius (Fig. 1f). Finally, droplets with higher surface tension pull more strongly on the ferrofluid, resulting in larger volume (and hence the magnetization) of the wetting ridge. As a result, higher surface tension also led to an increased average sliding velocity (Fig. 1g). Note that the high absolute drop velocities up to 5 mm/s were facilitated by the excess of ferrofluid on the sample, which guarantees the formation of a fully developed wetting ridge with minimum viscous dissipation[36].

**Versatility of WRAP**. The above results suggest that WRAP can actuate droplets both individually (Fig. 1c, e–g) and collectively (Fig. 1a). In addition, WRAP offers a range of unique advantages. First, the indirect actuation via the wetting ridge allows to manipulate a wide range of fluids, from pure water, to organics (Fig. 1g), and biological droplets with complex composition. In Fig. 2a–c, we demonstrate the actuation of Dulbecco's modified Eagle medium (DMEM, a widely used basal medium to support cell growth, Fig. 2a), fetal bovine serum (FBS, Fig. 2b), and buffy coat (Fig. 2c). Second, WRAP can actuate solid samples such as dried E.coli samples (Fig. 2d, e) or synthesized chemical particles (Supplementary Fig. 7), since solid samples also naturally raise wetting ridges (Fig. 2d). This is a function that is fundamentally impossible for many other droplet actuation methods. Third, WRAP can be operated on variable substrate geometries, such as straight (Fig. 2f, Supplementary Video 3) and curved (Fig. 2g, Supplementary Video 4) channels (made of glass capillary) as well as outside of a cylindrical (glass) rod (Fig. 2h, Supplementary

Video 5). Moreover, the strength of the magnetic force is sufficient to actuate droplets against gravity, as demonstrated in Fig. 2i and Supplementary Fig. 8 for 45° tilted surfaces, and a vertically oriented one in Fig. 2j (Supplementary Video 6). Fourth, WRAP also allows for programmable actuation. As shown in Fig. 2k, a droplet (dyed for later fluorescence characterizations) can be actuated along complex paths (such as a letter 'M') by sequentially activating electromagnets at different locations beneath the surface (Supplementary Video 7). Fifth, during all these operations, the anti-fouling properties of the liquid-infused surface that have been demonstrated in many previous studies[37,38] are fully preserved: no traces of residual fluorescence could be detected after manipulating the dyed droplet along a complex path (Fig. 2l). Finally, the choice of the ferrofluid is also versatile. For example, to actuate low surface tension droplet, fluorocarbon-based ferrofluid can be used (Supplementary Fig. 9). All these results suggest that our actuation method may be employed in biomedical analysis involving complex fluids without suffering from fouling, on both closed (Fig. 2f, g) and open microfluidic (Fig. 2a–c, h, j) platforms, and can be operated programmably (Fig. 2k).

**Discrete actuation of droplets and its application**. The capability of WRAP to actuate individual droplets in a discrete and programmable manner allows us to implement more complex digital microfluidic droplet operations such as droplet splitting, droplet generation, droplet transport, and merging of droplets. The latter two are easy for WRAP, but the first two are less clear. Since splitting a droplet automatically generates smaller droplets, droplet generation and splitting can be realized in the same way. Splitting droplets by magnetic forces has been a challenging task, especially in the absence of magnetic additives within the droplet[29]. Here, we designed a "ferrofluid-based cutting" technique to split the droplet based on the magnetically induced protrusions at the ferrofluid surface, as illustrated and experimentally demonstrated in Fig. 3a–d (see also Supplementary Video 8). The procedure involves several steps: first, we activated a magnet in Position 1 (Fig. 3a) to pull part of our reservoir droplet into a channel, a geometric constriction on the surface, until that channel was completely filled with the droplet fluid (Fig. 3b). Then, a magnet at Position 2, close to the upstream entrance of the channel was activated to create a protrusion of ferrofluid above the magnet (Fig. 3c). This cut the droplet in the channel into two parts. We then moved the upper part out of the channel by actuating a magnet in Position 3 (Fig. 3d). Here the electromagnet was placed further than the electromagnet 2 to induce smaller actuation force, so that the bigger droplet was not trapped again. Theoretically, the radius of generated droplets has a positive correlation with the volume of the channel. Therefore, as demonstrated in Fig. 3e and Supplementary Fig. 13, the radius can be controlled by varying the channel length or height.

In Fig. 2 we demonstrated that complex transport paths can be achieved by sequentially activating magnets at different locations (or equivalently by moving activated magnet). More elegantly, complex paths can be achieved by incorporating a patterned ferromagnetic layer into the substrate below the liquid-infused surface. Such a layer, which can be fabricated in various ways (see Methods and Supplementary Fig. 10), modulates the magnetic field distribution (simulated horizontal projection of the magnetic flux density in Fig. 3f. Details of the simulation in Methods and Supplementary Fig. 11) and guides droplets along paths imposed by the pattern. Droplets can be forced to move along curved paths (Fig. 3g, Supplementary Video 9). Or, in path with Y-shaped junctions droplets originating from different sources can be brought together and merged using a single actuator (Fig. 3h and

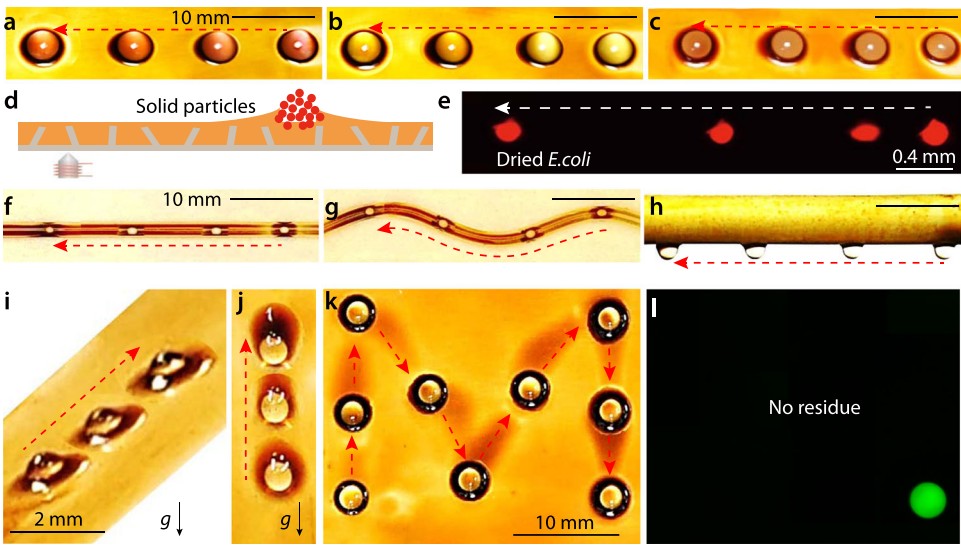

**Fig. 2 Versatility of the actuation. a–c** Actuation of biological droplets of Dulbecco's modified Eagle medium (DMEM, **a**), fetal bovine serum (FBS, **b**), and buffy coat (**c**). **d, e** Schematic illustration and fluorescence image showing the actuation of solid samples (dried *E.coli* cells). *E.coli* was dyed with propidium iodide (a dye that visualizes dead cells). **f, g** Actuation of a droplet inside a straight (**f**) or curved (**g**) tube with the inner surface infused with ferrofluid, which is useful in closed microfluidics. **h** Actuation of a droplet hanging upside down on the rod with the outer surface infused with ferrofluid. **i, j** On an open surface, the droplet could be transported against gravity, either when the surface was tilted by 45° (**i**) or 90° (**j**). The black arrows and the symbol *g* indicate the direction of the gravitational force. **k** By programmed sequential actuation of neibhourghing electromagnets, the droplet was guided to walk along a M-shape path. Electromagnets were beneath the droplets' snapshots. The droplet was dyed with fluorescent molecules. **l** The fluorescence image of the final state in **k** did not show any residue fluorescence on the walked path, suggesting the anti-fouling property. Dotted arrows indicate the moving direction of the actuated droplet.

Supplementary Fig. 10). Furthermore, the field modulation pattern can be conveniently switched to guide sequential droplet actuation (Supplementary Fig. 12). To demonstrate the possibility of biochemical analysis using this unique capability of WRAP, we performed a Griess test (Fig. 3g) which is widely used in diagnosing urinary tract infections[39] and detecting forensic nitroglycerine[40]. Droplets of Griess reagent and sodium nitrite solution from different reservoirs were transported and merged to a sample analysis region, which was divided into three sections to analyze different samples spectroscopically. Finally, the merged droplet was transported to the waste deposit region. In our experiments, we used three nitrite samples with initial concentrations of 0, 0.4, and 0.9 mM (concentrations down to 1 µM can also be analyzed, see Supplementary Fig. 14), respectively, and analyzed the samples merged with Griess reagent with UV-vis spectrometer (Fig. 3j). These results demonstrated the potential of WRAP in digital droplet microfluidics.

**Collective droplet actuation and its applications**. While discrete actuation of droplets is important in droplet microfluidics, the collective actuation of many droplets, especially micro-sized droplets, is crucial in droplet condensation, water collection, and potentially the detection of pathogens such as bacteria and viruses from respiratory droplets. The latter is believed to play a central role in the transmission of the coronavirus and the related Covid-19 disease[41]. In Fig. 1a we demonstrated the capability of WRAP to simultaneously actuate an ensemble of microdroplets. This capability can be further combined with ferromagnetic field modulation layers, formed with iron needle arrays, to collect droplets at specific locations on a surface. Figure 4a–e shows the accumulations of sprayed microdroplets in a square lattice (Fig. 4b, c) and an N-shape lattice (Fig. 4d, e, Supplementary Video 10).

The capability of collecting small droplets at well-defined locations is of particular interest if it enables the detection of

pathogens that would otherwise be undetectable due to low signal-to-noise ratios. For respiratory droplets, a particular challenge is that they display a broad size distribution including many small droplets of a few micrometers. Once deposited on a solid surface, such small droplets are difficult to manipulate because their large surface- and edge-to-volume ratio make droplets easy to be pinned on tiny surface defects. Moreover, the random positioning of droplets sprayed or coughed onto a surface precludes any actuation mechanism that relies on well-defined drop positions and/or sizes relative to actuator(s), such as electrodes. The unique capabilities of WRAP, however, allow for such actuation for a wide range of droplet sizes and arbitrary initial positions (Fig. 4f). In Fig. 4g, we show the accumulation of sprayed microdroplets (ranging from ~$10^1$ µm to hundreds of µm) containing dye-stained *E. coli* ($10^7$ colony-forming unit per mL, or CFU/mL) on a ferrofluid-infused slippery surface with a field modulation layer with square lattice to guide droplet accumulation. Without activating the magnet, randomly coalesced and distributed droplets were too small, and the fluorescence signal could be barely detected. Upon activating the magnet, however, the smaller droplets coalesced and accumulated at the pre-defined locations. This led to an almost 20-fold increase of the fluorescence signal that could easily be detected. By averaging the intensities on the lattice pattern, we can quantify the pathogen concentration. As shown in Fig. 4h, the fluorescence intensity increases linearly with the logarithm of the *E. coli* concentration, a phenomenon that happens at moderate analyte concentration[42,43].

For a successful detection of coronavirus or other pathogens from respiratory droplets, the collection device can be further coupled with a different detection mechanism and possibly with immunoassays or nucleic acid assays.

In summary, we developed a wetting ridge assisted programmable magnetic droplet actuation method. The method is applicable to droplets of various sizes and compositions, and to solid samples as well. Furthermore, its applications in digital

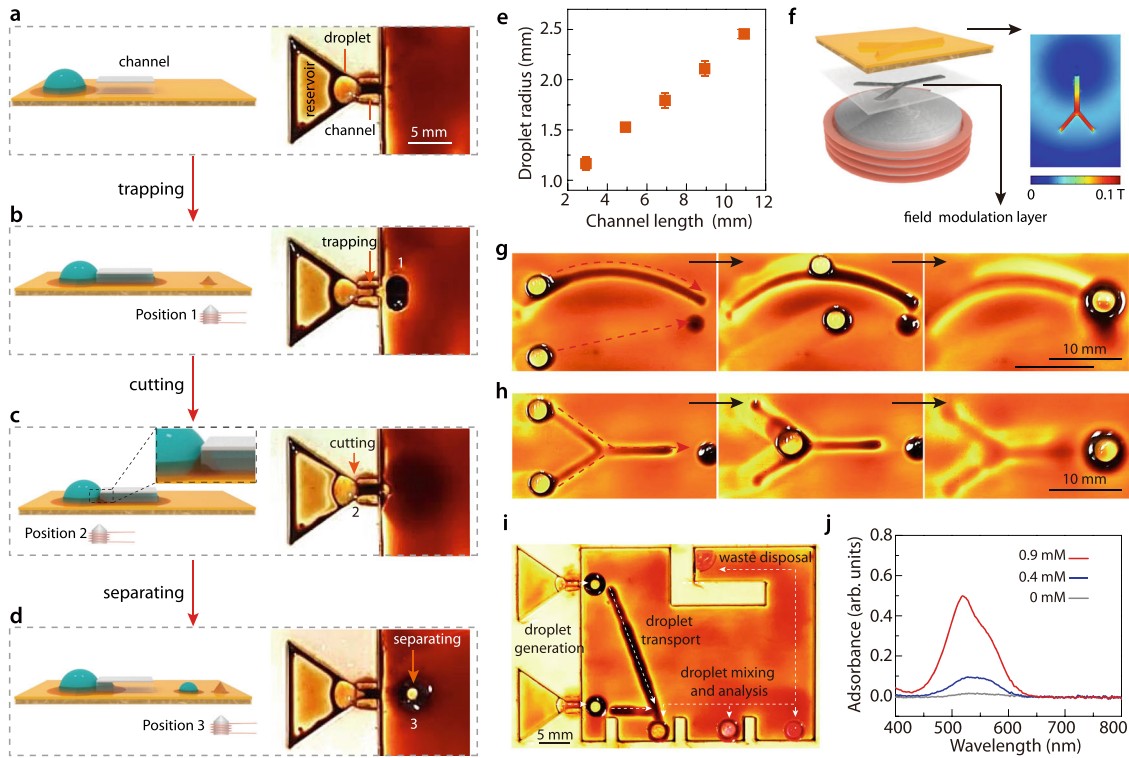

**Fig. 3 Discrete actuation of droplets for magnetic digital microfluidics. a–d** Schematic illustration (left schemes) and optical images (right) of the droplet generation/splitting process. A big droplet was placed outside a channel (**a**), and then trapped inside by actuating the electromagnet in Position 1 (**b**). After that, an electromagnet in Position 2 was actuated, raising a ferrofluid cusp (**c**), which cut the droplet (inset in the left scheme). Finally, an electromagnet in Position 3 was actuated to pull the smaller droplet out (**d**). **e** The size of the generated droplet increases with the channel length, showing promise for quantitative control. The error bars correspond to the standard deviations, calculated from at least three independent measurements. **f** To drive the droplet along a more complicated path, we can modulate the magnetic field by inserting a patterned paramagnetic layer. The resultant distribution of the horizontal projection of the magnetic flux density follows the paramagnet pattern (right inset). **g** Droplet placed on the pattern was therefore driven along the pattern, while the droplet placed away from the pattern was not. **h** A Y-shape pattern was used to guide droplet merging. **i** A proof-of-concept magnetic digital microfluidic chip, where droplets containing analytes were generated, transported, merged to induce chemical reaction, analyzed, and then deposited to the waste region. **j** UV-vis spectra of the reacted droplet.

microfluidics and bioanalysis were demonstrated. Future studies may be directed to quantify the coupling of hydrodynamics and magnetic dynamics in the droplet movement, such that higher precision actuation could be realized for a broader range of applications.

## Methods

**Fabrication of the ferrofluid-infused surface.** To fabricate such surface, we infused ferrofluid in a superhydrophobic aluminum substrate. The super-hydrophobic substrate was prepared using a reported method[44]. Briefly, an aluminum sheet ($50 \times 50 \times 0.5$ mm) was polished with sandpaper to remove the native oxide layer and then immersed in the deionized water at 95 °C for 1 h. As a result, a layer of nanostructured $Al_2O_3$ would form. Then, we hydrophobized the surface with 1H,1H,2H,2H-perfluorodecyltriethoxysilane (J&K) with vapor deposition method. This fluorosilane was deposited on the aluminum in a vacuum chamber and kept for 12 h at 80 °C. The structure of the superhydrophobic substrate was investigated with scanning electron microscope (Philips XL30CP, Supplementary Fig. 2a). The substrate exhibits a low contact angle of ~3° to silicon oil and a high contact angle of 161° to water (Supplementary Fig. 2b), suggesting that the superhydrophobic substrate favors the infusion of silicon oil. In Fig. 2f–g, the tube was made of glass capillary with ferrofluids infused on the inner surface. The curved tube was made by bending the capillary during heating.

The ferrofluid was fabricated following a previously reported recipe[16]. To synthesize $Fe_3O_4$ nanoparticles, aqueous solutions of $FeCl_3$ and $FeSO_4$ (J&K) were mixed with the $Fe^{3+}/Fe^{2+}$ molar ratio equaling 3:2. Then aqueous ammonia solution (28–30%, J&K) was added to adjust the pH value to >11, followed by adding carboxylate terminated silicone stabilizer (Gelest MCR-B12) to modify the nanoparticles. The mixture was vigorously stirred for 12 h. Then the nanoparticle was accumulated and held tightly with a magnet, and washed with acetone once and toluene twice. Finally, it was dispersed in the dimethyl silicone (12 cst) to obtain the silicon oil based ferrofluid. The concentration of $Fe_3O_4$ nanoparticles

can be controlled by tuning the mass of dimethyl silicone. Then the prepared ferrofluid was dip-coated on the superhydrophobic aluminum substrate, and the amount of ferrofluid was controlled such that its thickness is roughly tens of micrometers. The particle size distribution of $Fe_3O_4$ was measured with the dynamic light scattering (DLS) method using Malvern Zetasizer Nano-ZS ZEN3600. The dynamic viscosity measurements were performed on a rotational rheometer (Malvern Kinexus Lab+). Magnetic properties of the ferrofluid were measured with a vibrating sample magnetometer (Lakeshore). In the manuscript, the thickness ($d$) of the ferrofluid refers to the thickness of the overlayer. It was estimated by $d = m/\rho A$, where $\rho$ is the density of the ferrofluid, $m$ the mass of the ferrofluid, and $A$ the area of the substrate.

To synthesize fluorocarbon-based ferrofluid, aqueous solutions of $FeCl_3$ and $FeSO_4$ (J&K) with the $Fe^{3+}/Fe^{2+}$ molar ratio equaling 3:2, together with perfluoropolyether end-functionalized with a carboxylic acid group (PFPE-COOH, DuPont 157FSL), were mixed under magnetic stirring with a speed of 500 rpm/min at room temperature[45]. The m/v ratio of the generated $Fe_3O_4$ nanoparticles and PFPE-COOH was 2:1. Afterwards, aqueous ammonia solution (28–30%, J&K) was added and the reaction was allowed to proceed at 80 °C for an hour. Then the product was washed sequentially with water, ethanol, and water until pH = 7. The modified $Fe_3O_4$ nanoparticles were vacuum dried overnight at 80 °C, followed by re-dispersing in Krytox 100 to obtain the fluorocarbon-based ferrofluid.

**Discrete actuation of droplets.** Single microdroplets with controlled diameter were generated by the superhydrophobic syringe with a flat tip and an inner diameter of 60–1000 μm. The superhydrophobic needle was fabricated by coating fumed silica (22 nm, Sigma–Aldrich) on a steel needle. The superhydrophobicity reduces the adhesion of the droplets and improves the accuracy of droplet diameter. To actuate droplets using the WRAP method, a DC electromagnet (composed of an iron core and externally wrapped by copper wire loop, with a radius of 4 cm and a height of 8 cm. Detailed information can be found in Supplementary Fig. 11a. Magnetic field on top of the electromagnet is typically in the mT level) was placed beneath the slippery surface away from the droplets. Either permanent magnets or electromagnets can be used to actuate the droplets. While permanent

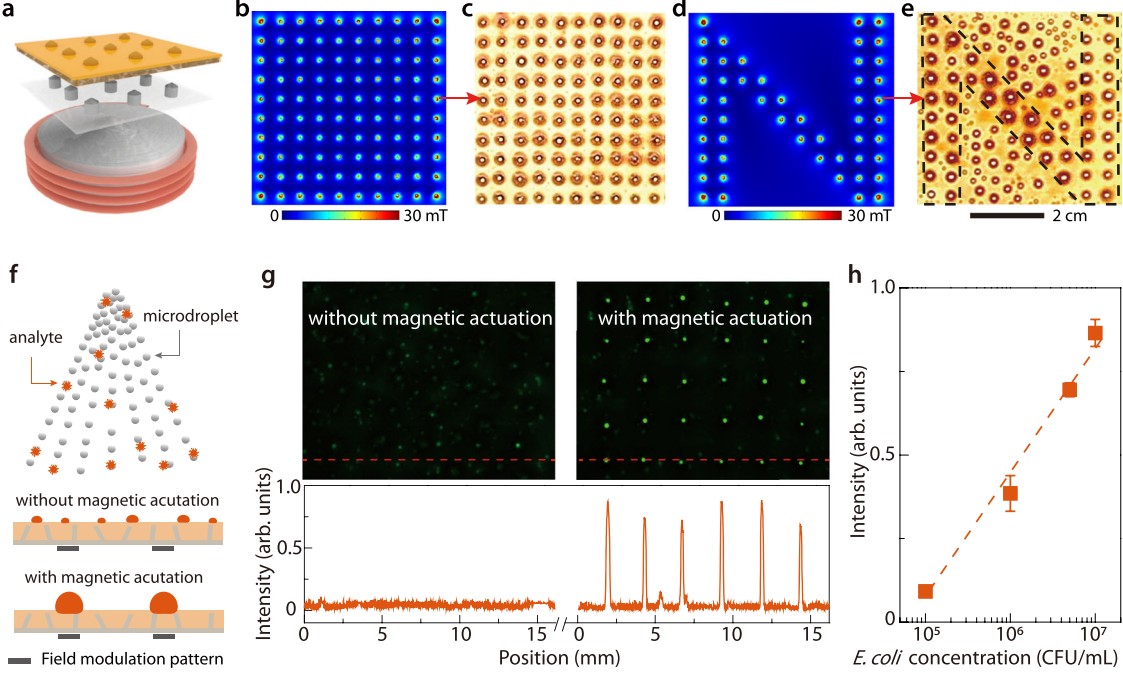

**Fig. 4 Collective actuation of microdroplets for droplet collection and biomedical analysis. a** A field modulation layer with a lattice pattern was used to guide the microdroplets coalescence. For a square lattice pattern, **b** the calculated magnetic flux density (horizontal projection) on the slippery surface also followed a square lattice pattern. **c** When micro-sized water droplets were sprayed on the surface, they coalesced and formed a square lattice pattern. **d**, **e** For a modulation layer with a N-shape lattice pattern, the calculated magnetic flux density (horizontal projection) shown in **d** and the coalesced microdroplets shown in **e** also followed a N-shape lattice pattern. Dashed lines in **e** are eye-guide. **f–h** This property can be used to collect respiratory droplets and detect the pathogens. **f** For a proof-of-concept demonstration, microdroplets containing dyed *E. coli* were sprayed on the surface with a square lattice pattern field modulation layer. **g** Without magnetic actuation, some microdroplets randomly coalesce, and showed fluorescence signals that were randomly distributed and were barely above noise level (left image). With magnetic actuation, all the microdroplets coalesce to form a lattice pattern, showing strong and regularly distributed fluorescence signals (right image). The bottom graph shows the fluorescence intensity along the dashed oranges lines on the images. **h** The fluorescence intensity increases almost linearly with the logarithm of the *E. coli* concentration, showing potential for quantification. Error bars represent the standard deviation ($n = 3$).

magnets typically provide stronger actuation force, we chose electromagnets here because they allow easier automation, better programmability, and higher compatibility with miniaturized platforms. The motion of the droplets was characterized with a Nikon Camera (D550) at 30 frames per second. In Fig. 1g, N, N-dimethylformamide was used to generate the lowest surface tension, and water was used to generate the highest surface tension. The others were generated by mixing ethylene glycol and water with varying volume ratios until the target surface tension was reached. The surface tensions were determined with DataPhysics contact angle system. In Fig. 2c, the buffy coat (10 μL) was isolated from human blood sample (Hongkong Red Cross) via centrifugation at 4000 rpm for 10 min. The experiment was carried out at BLS-2 lab. To programmably actuate droplets to walk along the M-shape line, nine electromagnets were placed beneath the designed line, and sequentially activated and deactivated. For this experiment, the droplet was dyed with Rhodamine B (J&K) and observed with Leica MZ10F Fluorescent Stereomicroscope to characterize the anti-fouling property of the substrate.

**Actuation of solid samples**. In this experiment, dried *E. coli* samples were actuated. Home-cultivated *E. coli* ($10^5$ colony-forming unit per mL, or CFU/mL) was used. To prepare the *E. coli* sample, a single colony of *E. coli* was cultured in Luria- Bertani broth at 37 °C for 12 h. The Luria-Bertani broth was prepared by adding 25 g broth powder (Invitrogen) in 1 L distilled water, followed by autoclaving at 121 °C for 20 min. Then *E. coli* was harvested centrifugally, washed with sterilized phosphate-buffered saline solution (PBS) for two times. Afterwards, the pellet was re-suspended with PBS and stored at 4 °C. For the experiment, 1 mL *E. coli* suspension ($10^5$ CFU/mL) was incubated with 1 μL red-fluorescent nucleic acid stains (propidium iodide (PI), Thermo Fisher Scientific) for twenty minutes. 0.06 μL (1 μL Microsyringe) dyed *E. coli* droplet was pipetted onto the surface and dried via evaporation in ambient condition. Electromagnet was adopted for the generation of external magnetic field. The actuation process was captured by Nikon Eclipse Ni-E upright fluorescence microscope.

Another actuated solid sample was ZIF-8 powder[46], a metal-organic-framework material that is frequently used in molecular separations, catalysis, and chemical analysis. To synthesize it, 34.9 mg ZrCl4 (J&K) and 24.9 mg 1-4-benzendicarboxylic acid (J&K) were dissolved in 10 mL N,N-Dimethylformamide (DMF), and the mixture was magnetically stirred at 120 °C for 12 h. After collected by

centrifugation at 9000 rpm/min, the particles were washed with DMF three times and vacuum dried overnight at 80 °C. Electromagnet was adopted for the generation of external magnetic field. The actuation process was recorded by a Nikon Camera (D550) at 30 frames per second.

**Imaging the wetting ridge**. The confocal images of the wetting ridge were captured with Zeiss LSM 880 laser confocal microscope at Z-stacking mode. The fluorescent Coumarin-6 (J & K Scientific) was dissolved in the dimethyl silicone (12 cst) by ultrasonic dispersion for 12 h, and dimethyl silicone served as the dispersion solution of ferrofluid.

**Interfacial tension measurement**. Optical images of pendant drop of various liquids at equilibrium were captured with KSV Instruments's CAM 101 system. Interfacial tension results were then calculated using ImageJ with a pendant drop analysis plug-in[47].

**Fabrication of the field modulation pattern**. The pattern can be fabricated in various ways. For the experiments in Fig. 3 in the main manuscript, we used a PDMS/ Fe3O4 pattern. Briefly, we mixed silicone precursor, curing agent (Sylgard 184, Dow Corning), and the prepared Fe3O4 nanoparticles at a weight ratio of 10:1:30 and stirred for 2 h. The mixture was poured on a home-carved mask and cured at 70 °C for 10 h to obtain the ferromagnetic field modulation pattern. The thickness of the mask and thus the pattern were 1 mm. Besides, one can also directly print PDMS/Fe3O4 pattern on the slippery surface to modulate the field (Supplementary Fig. 10). For the experiments in Fig. 4, an array of cylindrical iron needles (radius: 0.5 mm, height: 10 mm) was used to form the matrix pattern.

**Collective actuation of droplets**. The microdroplets were produced with a thin film chromatography sprayer (250 mL, Sigma–Aldrich). The distance between the sprayer and the substrate was set to be 10 cm. An electromagnet was placed beneath the center of the substrate, with or without a field modulation pattern in between, to actuate the droplets. The actuation process was recorded with a Nikon camera (D550) at 30 frames per second. For biomedical analysis, microdroplets containing dyed home-cultivated *E. coli* ($10^7$ CFU/mL) were used to simulate

respiratory droplets. To prepare the *E. coli* sample, a single colony of *E. coli* was cultured in Luria- Bertani broth at 37 °C for 12 h. The Luria-Bertani broth was prepared by adding 25 g broth powder (Invitrogen) in 1 L distilled water, followed by autoclaving at 121 °C for 20 min. Then *E. coli* was harvested centrifugally, washed with sterilized phosphate-buffered saline solution (PBS) two times. Afterwards, the pellet was re-suspended with PBS and incubated with green-fluorescent nucleic acid stains (SYTO 9, Thermo Fisher Scientific) for twenty minutes. After incubation, the dyed *E. coli* was collected by centrifugation and washed with PBS twice. The pellet was re- suspended with PBS and used as the medium for generating microdroplets. The dyed *E. coli* concentration in the suspension was estimated to be $10^7$ CFU/mL with serial dilutions and CFU count. The fluorescence signal of these microdroplets was analyzed on Leica MZ10F Fluorescent Stereomicroscope.

**Electromagnet field simulation**. Finite element analysis (COMSOL Multiphysics 5.4, MA, USA) was used to calculate the electromagnetic field. In the simulation setup, magnetic and electric field physics were used in an air environment. The electromagnet components and their dimensions (see Supplementary Fig. 11), and the electric current (direct current, 1.5A, 600-turn coils) were chosen as in the experiments.

**Griess test**. Bioassay of nitrite determination was performed using the Griess Reagent Kit (Invitrogen, Catalog number: G7921) which provides all the reagents required for nitrite quantitation. The Griess Reagent was formed by mixing equal volumes of N-(1-naphthyl) ethylenediamine and sulfanilic acid according to the product description. Deionized water was used to dilute nitrite standard solution (1.0 mL of 1.0 mM sodium nitrite in deionized water) to different concentrations. This Griess Reagent was deposited in one sample reservoir, and nitrite solutions of different concentration were deposited in the other sample reservoir. The final reaction system was composed of deionized water, nitrite solution and the Griess Reagent with a volume ratio of 26:3:1. After the Griess Reagent and nitrite solution merged and reacted for 30 min at room temperature (25 °C), the nitrite concentration was analyzed with UV-Vis spectrometer (NanoDrop 2000, Thermo Scientific).

**Acquisition of human blood**. We obtained the human blood sample directly from the Hong Kong Red Cross. The research was reviewed and approved by the College Human Ethics Sub-committee of City University of Hong Kong and conducted according to the principles of the Declaration of Helsinki. Informed consent was obtained from all donors and patients.

**Reporting summary**. Further information on research design is available in the Nature Research Reporting Summary linked to this article.

## Data availability

The data that support the findings of this study are provided within the paper and the supplementary information files. Any other relevant data are available from the corresponding author upon reasonable request.

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

## Acknowledgements

This work was supported by the Research Grant Council of Hong Kong (CityU 11305219), City University of Hong Kong (9667203), and Shenzhen Basic Research Program (JCYJ20180307123925200).

## Author contributions

J.Z. and X.W. contributed equally to this work. X.Y. and J.G. conceived the idea. X.Y., F.M., and J.G. supervised the project. J.Z., Z.W., X.W., B.Y., and L.A. conducted the experiments. B.Y. and L.A. contributed to fabrication and characterization of ferrofluids. Z.W. and X.W. contributed to bacteria detection. J.Z., X.W., Z.W., S.P., B.Y., L.A., J.G., F.M., and X.Y. contributed to data analysis and manuscript drafting.

## Competing interests

The authors declare no competing interests.
