## [Peer Review File · Nature Communications]

REVIEWER COMMENTS

Reviewer #1 (Remarks to the Author):

The paper "Wetting ridge assisted programmed magnetic actuation of droplets on ferrofluid-infused surface" is devoted to the important topic and presents valuable experimental findings. However, the paper needs a very deep revision.

1. It remains unclear for a reader: are droplets completely coated (cloaked) with the reported ferrofluid? What is the exact value of the spreading parameter describing the wetting situation reported in the manuscript. It is reasonable to suggest that droplets are completely coated with the silicone oil based ferrofluid, see:

Multanen V., et al., Hydrophilization of liquid surfaces by plasma treatment, *Colloids and Surfaces A*, 461 (2014) 225-230.

This point should be addressed under revision in detail.

2. If the droplets are completely coated by the silicone oil based ferrofluid, the disjoining pressure within thin layer of the oil becomes very important, see:

Rozenman Z., et al., Progress in low voltage reversible electrowetting with lubricated polymer honeycomb substrates. *RSC Adv.*, 2015,5, 32491-32496

These points should be addressed under the revision.

Reviewer #2 (Remarks to the Author):

The authors reported a programmable magnetic actuation of micron-sized non-magnetic droplets on ferrofluid-infused porous surfaces. The key advantage of this approach is the ability to collect and merge large numbers of droplets of different sizes over large areas, with potential applications in the diagnosis of droplets bearing COVID-19 pathogens. Moreover, the authors demonstrated droplet generation (splitting), droplet transport and merging, and fluorescent spectroscopic analysis of droplet content, which together form the basis of a rudimentary set of digital microfluidic operations. Overall, this manuscript represents a very important and timely contribution to the growing number of applications that employ ferrofluid-infused porous surfaces, and given its potential wide and immediate impact in digital microfluidics and disease diagnosis, it deserves to be published in *Nature Communications*. However, there are a number of issues that need to be resolved.

1. The authors use electromagnets extensively in this report. Yet, no specifications were given on the size and type of electromagnets, the field strengths and/or the distributions of the magnetic fields. Compared with NdFeB permanent magnets, the field generated by electromagnets is relatively weak. Could the authors comment on the pros and cons of using electromagnets vs permanent magnet for actuating droplets?

2. The authors use silicone-based ferrofluid for this study. This is a good choice because it allows the manipulation of water- and hydrocarbon-based pure and complex fluids. Nevertheless, the fluorocarbon-based ferrofluid could also be a good choice, especially given that the reagent used in the surface functionalization of the aluminium substrate is fluorocarbon-based molecules. Could authors comment why they do not consider using fluoro-carbon based ferrofluid?

3. In Figure 1a, the ferrofluid under the permanent magnet (what size and type?) do not show the characteristic surface instability (the so-called Rosensweig instability). It suggests either low field strength or low magnetization of the ferrofluid. Have the authors characterized the magnetic/fluidic properties of the ferrofluid they synthesized, such as M-H (B-H) curve, its saturation magnetization, and viscosity?

4. In Figure 1f, it is noted that the average velocity of the droplets increases with droplet size. How do the authors determine the droplet size? Should there be an error bar along the horizontal (droplet radius) direction too? How many data points were there for each data point? How do the authors rationalize the anomaly for sizes between 1.0 – 1.5 mm?

5. In Figure 2, what type of tube is used in the demonstration in panel a and b? They seem to be transparent. Are they still aluminium?

6. The droplet splitting experiment is particularly interesting. It could open up a new range of possibility for manipulating droplets on ferrofluid-infused surfaces. Have the authors considered the effect of channel height on the droplet volume?

7. The collective actuation for droplet collection demonstrates that merging small droplets could enhance the signal-to-noise ratio of fluorescent signals. This is a novel and useful strategy. Is there any particular significance of the N-shape used in panel d and e?

8. Some small suggestions:

- page 3, first paragraph: “arbitrary complexity”, not correct. Consider silicone-based droplets.
- page 4, last paragraph: “notoriously difficult”, is it true? Could the authors provide references?
- page 4, same paragraph: “very small surface- and edge-to-volume ratio”. Small or large?
- Methods: The authors need to add details to their experimental procedures so that others could reproduce their results.

Reviewer #3 (Remarks to the Author):

This paper presents a different way of actuating droplets on surfaces that uses a ferro-fluid- infused porous surface to allow for formation of a wetting ridge round a droplet and which carries a droplet in response to a remote magnetic field. This reviewer believes the basic concept of actuation is new and should be publishable as a fundamental contribution to science. However, the manuscript spends too much space trying to justify to the reader why this method is preferable to other digital microfluidic actuation, such as electrowetting-on-dielectric (EWOD). However, the authors have

failed to present arguments with any demonstrated basis that this is true. At least 6 companies now are selling EWOD based products for bio applications. (Lab Chip 2020 20 1705), and the present manuscript has not demonstrated that WRAP can do anything new beyond the EWOD state of the art, or for that matter, what EWOD research papers demonstrated 15 years ago. The one “unique” application proposed by the authors is to collect large amounts of drops with a random initial position and broad size distribution using a single actuator, thus enabling enhanced sensitivity in detecting airborne pathogens. However, the authors’ proposed WRAP collection method would have reduced signal-to-noise ratio, as explained below. Thus, while the science is good, the justifications for using WRAP in a practical way are unsupported and actually unconvincing. Thus, I do not support publication of this manuscript in its present form.

COMMENTS:

Abstract: Please drop the term “novel”. Readers will determine if this contribution is novel or not.

Line 33: “severely hindered”. Not true. In bio applications single use EWOD devices are required to prevent contamination from previous use. Multiple companies now offer successful products based on EWOD in the bio space and no instability issues are seen because the time of use is limited. (See, e.g. Kim et al, Lab Chip 2020 20 1705)

Line 50: authors emphasize that WRAP allows for actuation of droplets of water, organic fluids, as well as complex biological materials. However, EWOD has been shown to allow actuation of all these types of droplets. (See, e.g. Chatterjee et al, Lab Chip 2006 199 and Kim et al, Lab Chip 2020 20 1705)

Line 62: The authors identify the one unique aspect of WRAP: collect large amounts of drops with a random initial position and broad size distribution using a single actuator. They claim that such a capability “enables enhanced sensitivity in biomedical detection of airborne pathogens.” However, such a collection method does not allow for the target analyte to be concentrated for increased sensitivity. Rather, analyte dilution is likely in the collection droplet, reducing sensitivity.

Lines 146-155: authors demonstrate WRAP using a Griess test utilizing a solution of reagent and sodium nitrite. Absorbance was used to detect nitrite concentration above about 0.2mmol. This level is about 5-20 times larger than reported by Huang et al for airborne inorganic particle detection. And this level of detection seems much too large for many applications. For the applications in the paper, what detection limits are clinically important and are the results of Fig. 10 even close to being relevant?

Lines 165-184. WRAP collection of sprayed droplets does not allow detection of otherwise undetectable signal-to-noise ratios. The large WRAP collection droplet represents a larger source of noise, and the signal is determined by analyte concentration, which would be the same concentration as in a small droplet or less in the collection droplet due to dilution.

At line 188 the authors go on to claim that such collection of airborne pathogens is “...unmatched by other droplet manipulation methods.” They cite no references to support this claim. However, excellent airborne particle collection by scanning EWOD droplets has been demonstrated recently. (See, e.g. S. Huang, et al, Sensors, 20, 1281 (2020).) If this is the unique application of WRAP, the authors need to do a comparative analysis with the Huang paper. For example, by collecting particles in a single scanning droplet across a collection surface, Huang showed that the concentration of particles is enhanced for detection. In WRAP, the particle concentration is diluted in a large volume collection droplet, thus causing sample dilution rather than sample concentration.

Reviewer #1

The paper "Wetting ridge assisted programmed magnetic actuation of droplets on ferrofluid-infused surface" is devoted to the important topic and presents valuable experimental findings. However, the paper needs a very deep revision.

Comments 1): It remains unclear for a reader: are droplets completely coated (cloaked) with the reported ferrofluid? What is the exact value of the spreading parameter describing the wetting situation reported in the manuscript. It is reasonable to suggest that droplets are completely coated with the silicone oil based ferrofluid, see:

Multanen V., et al., *Hydrophilization of liquid surfaces by plasma treatment, Colloids and Surfaces A*, 461 (2014) 225-230.

This point should be addressed under revision in detail.

Response: We thank the reviewer for pointing out that we were indeed not very explicit with respect to cloaking in the original manuscript. In our droplet actuation system, droplets are indeed completely coated by ferrofluid, as illustrated in the schematic in Fig. R1-1 and experimentally shown in the optical image in Fig. R1-1 (ferrofluid is brown).

As suggested by the reviewer, we measured the interfacial tensions and calculated the spreading parameter S , which are all >0 , confirming that the water droplet should be cloaked by the ferrofluid. Citing the mentioned reference (Ref 31), we accordingly revised the manuscript (first paragraph in the Actuation of droplets with WRAP section), "Together with the positive spreading pressure and the resulting cloaking layer of ferrofluid on the water^{31,32} (see supplementary Table 1), this excess ensures that saturated wetting ridges form independently around each drop³³"

The table was included in the Supplementary Information.

Figure R1-1 | Water droplet (5 μ L) is cloaked by ferrofluid (brown).

Supplementary Table 1 Initial spreading coefficients

Interface 1-2 (γ_{12}) (mN/m)	Interface 3-1 (γ_{31}) (mN/m)	Interface 3-2 (γ_{32}) (mN/m)	Initial spreading coefficients ($S_i = \gamma_{12} - (\gamma_{31} + \gamma_{32})$)
water-air (72.05 \pm 0.31)	Dimethicone-air (17.20 \pm 0.18)	Dimethicone-water (20.34 \pm 0.33)	>0
water-air (72.05 \pm 0.31)	1%* ferrofluid -air (10.57 \pm 0.29)	1%* ferrofl.-water (23.28 \pm 0.41)	>0
water-air (72.05 \pm 0.31)	5%* ferrofluid -air (10.36 \pm 0.23)	5%* ferrofl.- water (22.94 \pm 0.36)	>0

water-air	12%* ferrofluid-air	12%* ferrofl.- water	>0
(72.05±0.31)	(8.33±0.25)	(19.29±0.59)	

Interfacial tension was calculated with pendant drop method. The concentration of ferrofluid refers to the volumetric concentrations of Fe₃O₄ nanoparticles. Note: in our experiments except Fig. 1e, 12% ferrofluid was used.

Comments 2): *If the droplets are completely coated by the silicone oil based ferrofluid, the disjoining pressure within thin layer of the oil becomes very important, see: Rozenman Z., et al., Progress in low voltage reversible electrowetting with lubricated polymer honeycomb substrates. RSC Adv., 2015,5, 32491-32496 These points should be addressed under the revision.*

Response: We are grateful to the reviewer's comment. Indeed, when the ferrofluid coats the droplet, the disjoining pressure should indeed become very important. Together with the pressure in the ferrofluid wedges, it controls the thickness of the cloaking layer and thereby affects the apparent contact angle, as discussed in *RSC Adv., 2015,5, 32491*. This coupling between disjoining pressure and wedge shape could indeed make the quantitative description of the magnetic driving force more difficult. However, as long as the ferrofluid wetting ridges exist (which is clearly visible in the experiments), the actuation can still proceed following the scenario in our manuscript.

We briefly discussed the cloaking layer in the revised manuscript (see Response to Comment 1), citing the mentioned reference (Ref 32)

Reviewer #2

The authors reported a programmable magnetic actuation of micron-sized non-magnetic droplets on ferrofluid-infused porous surfaces. The key advantage of this approach is the ability to collect and merge large numbers of droplets of different sizes over large areas, with potential applications in the diagnosis of droplets bearing COVID-19 pathogens. Moreover, the authors demonstrated droplet generation (splitting), droplet transport and merging, and fluorescent spectroscopic analysis of droplet content, which together form the basis of a rudimentary set of digital microfluidic operations. Overall, this manuscript represents a very important and timely contribution to the growing number of applications that employ ferrofluid-infused porous surfaces, and given its potential wide and immediate impact in digital microfluidics and disease diagnosis, it deserves to be published in *Nature Communications*. However, there are a number of issues that need to be resolved.

Comments 1): The authors use electromagnets extensively in this report. Yet, no specifications were given on the size and type of electromagnets, the field strengths and/or the distributions of the magnetic fields. Compared with NdFeB permanent magnets, the field generated by electromagnets is relatively weak. Could the authors comment on the pros and cons of using electromagnets vs permanent magnet for actuating droplets?

Response: We thanks the reviewer for this comment. Our DC electromagnet is composed of an iron core and externally wrapped by copper wire loop. The electromagnet is cylindrical, with a height of 8 cm and a radius of 4 cm. The detailed shape and size of the electromagnet is also given in Supplementary Fig. 11a. The magnetic field strength generated on top of the center of the electromagnet is in the order mT, measured by HT 201 Gaussmeter (Hengtong magnetoelectricity, CO., LTD.).

The above information is added in the manuscript and Methods section.

Supplementary Fig. 11a | Shape and size of the used electromagnet.

Regarding the pros and cons of electromagnet, indeed, permanent magnets indeed typically provide stronger magnetic fields and hence stronger actuation forces and higher droplet transport velocity, as we also explored in various test experiments. For our experiments in the present manuscript, however, we chose electromagnets because they allow easier automation and better programmability. In addition, electromagnets could be miniaturized down to microscale, making them easier to be integrated in miniaturized platforms.

We included the above discussions in the Methods section (see section: Discrete actuation of droplets), *“Either permanent magnets or electromagnets can be used to actuate the droplets. While permanent magnets typically provide stronger actuation force, we chose electromagnets here because they allow easier automation, better programmability, and higher compatibility with miniaturized platforms. Upon turning on an electromagnet underneath...”*

Comments 2): *The authors use silicone-based ferrofluid for this study. This is a good choice because it allows the manipulation of water- and hydrocarbon-based pure and complex fluids. Nevertheless, the fluorocarbon based ferrofluid could also be a good choice, especially given that the reagent used in the surface functionalization of the aluminum substrate is fluorocarbon-based molecules. Could authors comment why they do not consider using fluorocarbon-based ferrofluid?*

Response: We are grateful for this comment. We initially chose silicone-based ferrofluid for proof-of-concept demonstration because it has low cost. But of course, any ferrofluid can be used as long as it completely wets the substrate, is immiscible with the actuated droplet, and has lower surface tension than the droplet fluid.

Below we show that fluorocarbon-based ferrofluid was used to actuate silicone oil droplet (Supplementary Fig. 9). This result was included in the revised Supplementary Information to demonstrate the versatility of our approach, and briefly mentioned in the main text (Versatility of WRAP section), *“Finally, the choice the ferrofluid is also versatile. For example, to actuate low surface tension droplet, fluorocarbon-based ferrofluid could be used (Supplementary Fig. 9). All these results...”*

Supplementary Fig. 9 | Actuation of silicone oil droplet on fluorocarbon-based ferrofluid. (a) Side view of droplet movement along a straight line. (b) Programmed actuation of droplets along a M-shape path Scale bars in both images are 2 mm.

Comments 3): In Figure 1a, the ferrofluid under the permanent magnet (what size and type?) do not show the characteristic surface instability (the so-called Rosensweig instability). It suggests either low field strength or low magnetization of the ferrofluid. Have the authors characterized the magnetic/fluidic properties of the ferrofluid they synthesized, such as M - H (B - H) curve, its saturation magnetization, and viscosity?

Response: Thanks for the comment. In Figure 1a, a DC electromagnet was used, as described in the Response to Comment 2.

The magnetization of the ferrofluid was measured (Supplementary Fig. 3b) and the saturation magnetic susceptibility is 6.2 emu/g.

The surface instability indeed originates from the low magnetic susceptibility and the low magnetic field strength. The surface instability is determined by the competition between the Laplace pressure of the droplet which tends to keep the droplet shape, and the magnetic pressure which tends to deform the droplet. Thus, the “magnetocapillary” number (Mc), *i.e.* the ratio of magnetic pressure to Laplace pressure, $Mc = \frac{B^2}{\mu} R/2\gamma$ can be used to access this competition. Here B is the magnetic field, μ the magnetic permeability, R the ferrofluid droplet radius, and γ the ferrofluid/air interfacial tension. μ is related the magnetic susceptibility χ by $\mu = \mu_0(1 + \chi)$. Mc is therefore calculated to be around 10^{-2} - 10^{-1} , meaning that the Laplace is dominating, explaining why surface instability was not observed.

We also characterized the particle size distribution of Fe_3O_4 nanoparticles, which show an average size of 33 nm (Supplementary Fig. 3a). The ferrofluid has a low viscosity of ~ 0.01 Pa·S. (Supplementary Fig. 3c, the ferrofluid used for actuation has a concentration of 12%).

We included these results in the revised Supplementary Information.

Supplementary Fig. 3 | Properties of the ferrofluid used in the experiment. (A) The particle size distribution of Fe₃O₄ nanoparticles. (B) Magnetization curve of the used ferrofluid (12 v/v% Fe₃O₄ nanoparticles). (c) Viscosity of the ferrofluid in response to the volumetric concentration of the Fe₃O₄ nanoparticle.

Comments 3): In Figure 1f, it is noted that the average velocity of the droplets increases with droplet size. How do the authors determine the droplet size? Should there be an error bar along the horizontal (droplet radius) direction too? How many data points were there for each data point? How do the authors rationalize the anomaly for sizes between 1.0 – 1.5 mm?

Response: Thanks for these comments. In our experiments, droplets were produced from a superhydrophobic syringe with a flat tip. The inner diameter of the syringe was changed from 60 to 1000 μm to generate droplets of varying sizes. From the optical images of the droplet hanging down the syringe, we can calculate the droplet radius. We now added the error bar in the horizontal direction (see new Fig. 1f). For each data points, at least three independent measurements were done.

We realize that the droplet transport velocity for sizes between 1.0-1.5 mm is abnormal. There was some difficulty in this experiment. The actuation process is quite complex because the ferrofluid wetting ridge, which determines the actuation force, changes its shape along the transport path, making the actuation force changing all the time. Minor error in the control of the experimental condition would lead to large error in the data. We therefore carefully reviewed our experiment and repeated it for several times. The new data now shows that the average velocity increases monotonically with the droplet radius (see new Fig. 1f).

We would like to emphasize that these changes do not affect our main conclusion of this work in any way.

Figure 1f | The velocity increases with the droplet radius, presumably because the ridge volume increases with the droplet size. Error bars represent the standard deviation ($n \geq 3$).

Comments 4): In Figure 2, what type of tube is used in the demonstration in panel a and b? They seem to be transparent. Are they still aluminum?

Response: The used tube is glass capillary. This information is now provided in the revised manuscript and Method section.

Comments 5): The droplet splitting experiment is particularly interesting. It could open up a new range of possibility for manipulating droplets on ferrofluid-infused surfaces. Have the authors considered the effect of channel height on the droplet volume?

Response: Thanks for the suggestion. Indeed, the droplet volume can also be tuned by controlling the channel height. To demonstrate this, we changed the channel height and observed that the droplet radius increased with the channel height (Supplementary Fig. 13). This has been added in the revised manuscript in Paragraph 1 on Page 4, and in the Supplementary Information.

Supplementary Fig. 13 | Control of the droplet radius by varying the channel height. Fixed channel length: 3 mm. Fixed width: 1 mm.

Comments 6): *The collective actuation for droplet collection demonstrates that merging small droplets could enhance the signal-to-noise ratio of fluorescent signals. This is a novel and useful strategy. Is there any particular significance of the N-shape used in panel d and e?*

Response: Thanks for your question. N-shape was used as a demonstration, without any particular significance. It could be any other arbitrary shape.

Comments 7): *Some small suggestions:*

- *page 3, first paragraph: “arbitrary complexity”, not correct. Consider silicone-based droplets.*

- *page 4, last paragraph: “notoriously difficult”, is it true? Could the authors provide references?*

Response: Thanks for pointing out. We deleted “arbitrary” and “notoriously” in the revised manuscript

- *page 4, same paragraph: “very small surface- and edge-to-volume ratio”. Small or large?*

Response: Thanks for pointing this out. It should be large, and we revised it.

- *Methods: The authors need to add details to their experimental procedures so that others could reproduce their results.*

Response: Thanks for the suggestion. We added more details such that the results can be more easily reproduced. See changes highlighted in red in the revised manuscript.

Reviewer #3 (Remarks to the Author):

This paper presents a different way of actuating droplets on surfaces that uses a ferro-fluid- infused porous surface to allow for formation of a wetting ridge round a droplet and which carries a droplet in response to a remote magnetic field. This reviewer believes the basic concept of actuation is new and should be publishable as a fundamental contribution to science. However, the manuscript spends too much space trying to justify to the reader why this method is preferable to other digital microfluidic actuation, such as electrowetting-on-dielectric (EWOD). However, the authors have failed to present arguments with any demonstrated basis that this is true. At least 6 companies now are selling EWOD based products for bio applications. (Lab Chip 2020 20 1705), and the present manuscript has not demonstrated that WRAP can do anything new beyond the EWOD state of the art, or for that matter, what EWOD research papers demonstrated 15 years ago. The one “unique” application proposed by the authors is to collect large amounts of drops with a random initial position and broad size distribution using a single actuator, thus enabling enhanced sensitivity in detecting airborne pathogens. However, the authors’ proposed WRAP collection method would have reduced signal-to-noise ratio, as explained below. Thus, while the science is good, the justifications for using WRAP in a practical way are unsupported and actually unconvincing. Thus, I do not support publication of this manuscript in its present form.

Response: We thank the reviewer for his/her comments. This study is a proof-of-concept demonstration of WRAP, showing its potential to address some limitations of previous methods such as additive-assisted magnetic actuation, light-driven actuation, EWOD, and electro-dewetting. Overall, we agree with the reviewer to focus more on the fundamental science and strengths of WRAP and tune down the explicit comparison to other approaches. Accordingly, we removed the explicit reference to EWOD from the abstract and we reduced reference to EWOD throughout the introduction and main text in the revised manuscript.

However, we would like to point out that, from fundamental science perspective, WRAP does stand out as a promising alternative droplet actuation method. While one of us has been rather closely involved in the development EWOD for more than two decades and is also co-founder of one of the bio-application spin-off companies in the mentioned publication (LabChip 2020, 20, 1705), we feel that EWOD did not really fulfill all the expectations from 20 years ago, as illustrated by the withdrawal of the two largest investors (Illumina, Amazon) in recent years. This suggests that there are inherent problems that leave room for exploring alternative approaches. This challenge is apparently also shared by the main author of the suggested reference LabChip 2020, 20, 1705, who recently developed Electro-dewetting (Li et al. Nature, 572, 507–510, 2019; ref. 25 of our ms.) as an alternative technology in order to overcome what is described there as the fundamental unsolved challenges of EWOD. (Two quotes from that work: “... dielectric layer and a hydrophobic topcoat ... comprise reliability owing to dielectric breakdown, electric charging, and biofouling”;

“... (not having the) dielectric layer nor the hydrophobic topcoat, ... may avoid the reliability problems of EWOD.”)

By demonstrating drop motion, splitting, merging, we prove that WRAP can perform all basic drop operations. Yet, first, by definition WRAP does so without electric field-induced substrate degradation and hydrophobic fouling that are at the origin of most problems companies have experienced with EWOD devices and second, the additional flexibility of a long range driving force that does not require pre-defined electrode geometries enables additional flexibility such as collection of multiple random drops. This goes beyond EWOD, also beyond the approach of Huang et al. (See Fig. 1a)

In addition, the comment of the reviewer also triggered us to explore the actuation of dried biological samples (bacteria) and synthesized chemical samples (metal-organic-framework powder). These additional experiments turned out to be successful and demonstrate that WRAP simultaneously enables the manipulation of liquid droplets and solid samples (see below), which is fundamentally impossible for most other droplet handling techniques. This unique capability further demonstrates the high flexibility of WRAP and enhances the potential of WRAP even more. Obviously, this should enable new protocols for pre-concentrating particularly very dilute samples beyond the capabilities of most liquid droplet handling techniques, and is expected to be easier to operate than using a spanning droplet as in Huang’s work.

These results were included in Fig. 2 in the main manuscript and Supplementary Fig. 7 in the SI.

Figure R3-1 | WRAP can actuate both droplet and solid samples. (a) Fluorescence images showing the actuation of a droplet containing *E. coli.*, (b) a dried (solid, as indicated by the irregular shape) *E. coli.* sample, and (c) the simultaneous actuation of the droplet and dried samples. All scale bars are 0.4 mm.

Supplementary Fig. 7 | Actuation of metal-organic-framework (MOF) powder, showing that WRAP can actuate solid samples. (a) SEM image of the synthesized MOF. (b) Optical images of the actuation process

Comments 1): *Abstract: Please drop the term “novel”. Readers will determine if this contribution is novel or not.*

Response: Thanks for your comment. It has been dropped.

Comments 2): *Line 33: “severely hindered”. Not true. In bio applications single use EWOD devices are required to prevent contamination from previous use. Multiple companies now offer successful products based on EWOD in the bio space and no instability issues are seen because the time of use is limited. (See, e.g. Kim et al, Lab Chip 2020 20 1705)*

Response: Thanks for your comment. We dropped the term “severely”. Indeed, lifetime limit is not a problem for single-use EWOD devices, but it remains a problem for many other EWOD devices (see also ref. 25).

Comments 3): *The authors emphasize that WRAP allows for actuation of droplets of water, organic fluids, as well as complex biological materials. However, EWOD has been shown to allow actuation of all these types of droplets. (See, e.g. Chatterjee et al, Lab Chip 2006 199 and Kim et al, Lab Chip 2020 20 1705)*

Response: Response: Thanks for your comment. (see also general reply above) In this part of the demonstration, we did not compare our droplet actuation method to EWOD but proved the feasibility of our magnetic fluid manipulation platform for driving standard droplet operations. Moreover, our new experiments with dried bacteria and MOF powder also demonstrate the actuation of solid samples without droplet – a capability that is physically impossible for EWOD. We adapted the following accordingly (Page 3, paragraph 2):

“The above results suggest that WRAP can actuate droplets both individually (Fig. 1c, e-g) and collectively (Fig. 1a). In addition, WRAP offers a range of unique

advantages. First, the indirect actuation via the wetting ridge allows to manipulate a wide range of fluids, from pure water, to organics (Fig. 1g), and to biological droplets with complex composition.”

Comments 4): Line 62: The authors identify the one unique aspect of WRAP: collect large amounts of drops with a random initial position and broad size distribution using a single actuator. They claim that such a capability “enables enhanced sensitivity in biomedical detection of airborne pathogens.” However, such a collection method does not allow for the target analyte to be concentrated for increased sensitivity. Rather, analyte dilution is likely in the collection droplet, reducing sensitivity.

Response: Thanks for your comment. Indeed, we are not arguing that the concentration is increasing. What we point out is that the induced coalescence of many small drops (with the same concentration) into a bigger one increases the absolute signal. Since the major noise is the detection noise from the camera, the increase of the absolute signal results in the increase of signal-to-noise ratio. That’s what we show in Fig. 4 g-h.

Increasing concentration could be achieved by letting drops (small droplets or coalesced large droplets) partly evaporate. To demonstrate such, we observed the fluorescence signal of an evaporating PBS droplet (1 μ L) containing GFP-expressing *E. coli*. The results show that the signal increased significantly, as it can be seen from Fig. R3-2.

Furthermore, the ability of WRAP to handle even dried biological material enables maximum up-concentration.

Figure R3-2 | Increasing concentration by evaporating droplet. (a-b) Side-view optical image (a) and top-view fluorescence image of an evaporating droplet containing GFP-expressing *E. coli* (1×10^5 CFU). (c) Fluorescence intensity increases with time.

Comments 5): Lines 146-155: authors demonstrate WRAP using a Griess test utilizing a solution of reagent and sodium nitrite. Absorbance was used to detect nitrite concentration above about 0.2 mmol. This level is about 5-20 times larger than reported by Huang et al for airborne inorganic particle detection. And this level of detection seems much too large for many applications. For the applications in the

paper, what detection limits are clinically important and are the results of Fig. 10 even close to being relevant?

Response: Thanks so much for pointing this out. The concentrations showed in Fig. 3h were the initial concentration of the target. This has been specified in the Method section and in the revised manuscript.

The Griess test is only to demonstrate the droplet handling capability of WRAP and its potential in digital microfluidics. The limit of detection (LOD) was not a primary concern in this demonstration.

Nevertheless, we did a separate experiment to determine the LOD (Supplementary Fig. 14), and found that at least 1 μM nitrite can be detected, not worse than Huang's work.

It is worth mentioning that the fluorescence-based detection as such in WRAP is not unique compared to other drop-based Microfluidics platforms. What is unique is only the drop handling. Therefore, the sensitivity of fluorescence-based detection should be fundamentally the same independent of the droplet actuation method.

Supplementary Fig. 14 | Detection of low concentration nitrite. Nitrite with initial concentration ranging from 1 μM to 40 μM was also analyzed. The results show that 1 μM can quantitatively detected.

Comments 6): Lines 165-184. WRAP collection of sprayed droplets does not allow detection of otherwise undetectable signal-to-noise ratios. The large WRAP collection droplet represents a larger source of noise, and the signal is determined by analyte concentration, which would be the same concentration as in a small droplet or less in the collection droplet due to dilution.

Response: Thanks for your comment. For fluorescence microscopy, there are two main sources of noise (Technical documents from microscopy company: <https://www.olympus-lifescience.com.cn/en/microscope-resource/primer/techniques/confocal/signaltonoise/>), *i.e.*, Photon noise, and the electronic noise. The electronic noise originates from the instrument and can be considered constant. For this part, when we collect a large number of droplets, we dramatically increase the absolute

signal intensity, and therefore dramatically increases the signal-to-noise (SNR) ratio.

The poison noise is statistical variation in the number of detected photons, and is therefore determined by the number of photons received by the camera. The signal to noise ratio is given by, $SNR = \sqrt{N}$, where N is the photon number. Clearly, when we collect a large number of droplets, we dramatically increase N , and thereby SNR.

In either way, even if we do not increase the concentration by evaporation, we can dramatically increase SNR by collecting many small droplets.

Comments 7): At line 188 the authors go on to claim that such collection of airborne pathogens is “...unmatched by other droplet manipulation methods.” They cite no references to support this claim. However, excellent airborne particle collection by scanning EWOD droplets has been demonstrated recently. (See, e.g. S. Huang, et al, Sensors, 20, 1281 (2020).) If this is the unique application of WRAP, the authors need to do a comparative analysis with the Huang paper. For example, by collecting particles in a single scanning droplet across a collection surface, Huang showed that the concentration of particles is enhanced for detection. In WRAP, the particle concentration is diluted in a large volume collection droplet, thus causing sample dilution rather than sample concentration.

Response: Thanks for your comment. Again, we would like to point out that, for detection, WRAP is not unique compared to other drop-based microfluidics platforms. What is unique is the drop handling. Therefore, the resolution of fluorescence-based detection as such will be fundamentally the same as in other approaches, such as EWOD, which is also a drop handling platform only.

Here, “unmatched” refers to the capability of WRAP in easily collecting a large number of droplets from random initial positions with a wide distribution of sizes. There are, of course, many methods to collect droplets, an excellent example being EWOD. But they typically require more effort in device design and operation, and are typically more sensitive to the droplet size and position. For example, it is hard for EWOD to actuate a droplet that is much smaller than a single electrode. To actuate a large area of droplets, EWOD needs to sequentially actuate a large number of electrodes, necessitating effort in microfabrication and programming.

Regarding the concentration, WRAP collects droplets of the same initial concentration, and thus will not cause dilution. Huang’s work concentrated the analyte by pre-evaporate the tiny droplets, followed by collecting them with a spanning droplet. We, of course, can also concentrate the analyte *via* evaporation (Response to Comment 4). In addition, WRAP can do the same spanning (see programmed droplet actuation in Fig. 2i).

In particular, the unique actuation mechanism of WRAP even allows us to actuate dried solid analyte without using a spanning droplet, because solid objects can also raise ferrofluid wetting ridges. This allows to achieve maximum concentration.

Nevertheless, we agree with the reviewer’s previous general comment that focusing on the fundamental science and strength of WRAP is more appropriate. As a result, we dropped this statement in the revised manuscript.

REVIEWERS' COMMENTS

Reviewer #1 (Remarks to the Author):

The paper is interesting and it is publishable after the minor revision.

The English should be very deeply edited.

Remarks:

1. In the Text:

"Together with the positive spreading pressure and the resulting cloaking layer of ferrofluid on the water^{31,32} (see supplementary Table 1), this excess ensures that saturated wetting ridges form independently around each drop³³."

It should be:

"Together with the positive spreading parameter and the resulting cloaking layer of ferrofluid on the water^{31,32} (see supplementary Table 1), this excess ensures that saturated wetting ridges form independently around each drop³³."

2. The English should be edited.

Legend To Figure 4c. In the text:

"When microsized water droplets were sprayed on the surface, they coalesce and formed a square. lattice pattern."

It should be:

When microsized water droplets were sprayed on the surface, they coalesced and formed a square. lattice pattern.

Legend to Figure 4g. In the text: "With magnetic actuation, all the microdroplets coalesce to form a lattice pattern, showing regularly distributed and strong fluorescence."

It should be: "With magnetic actuation, all the microdroplets coalesce to form a lattice pattern, showing regularly distributed and strong fluorescence".

3. Figures 3b,e,f, Figure 4 need scale bars.

Reviewer #2 (Remarks to the Author):

The authors have extensively revised the manuscript. The new supplementary figures and tables substantiated the original statements. With all the additions and revisions, the authors have addressed all my concerns satisfactorily. This reviewer only has one more additional question. The

authors mentioned that the thickness of the ferrofluid layer is “roughly tens of micrometers”. How did the authors estimate/measure the thickness? Is this value the thickness of the overlayer only? Or does it also include the thickness of the porous substrate?

Response letter

Reviewer #1 (Remarks to the Author):

The paper is interesting and it is publishable after the minor revision.

The English should be very deeply edited.

Remarks:

1. In the Text:

"Together with the positive spreading pressure and the resulting cloaking layer of ferrofluid on the water^{31,32} (see supplementary Table 1), this excess ensures that saturated wetting ridges form independently around each drop³³."

It should be:

"Together with the positive spreading parameter and the resulting cloaking layer of ferrofluid on the water^{31,32} (see supplementary Table 1), this excess ensures that saturated wetting ridges form independently around each drop³³."

2. The English should be edited.

Legend To Figure 4c. In the text:

"When microsized water droplets were sprayed on the surface, they coalescence and formed a square. lattice pattern."

It should be:

When microsized water droplets were sprayed on the 491 surface, they coalesced and formed a square. lattice pattern.

Legend to Figure 4g. In the text: "With magnetic actuation, all the microdroplets coalescence to form a lattice pattern, showing regularly distributed and strong fluorescence."

It should be: "With magnetic actuation, all the microdroplets coalesce to form a lattice pattern, showing regularly distributed and strong fluorescence".

Response to 1 and 2: Thank you very much for the positive comment! We appreciate the additional remarks and have made corresponding revisions.

3. Figures 3b,e,f, Figure 4 need scale bars.

Response: Thanks for the suggestion! Scale bars have been added.

Reviewer #2 (Remarks to the Author):

The authors have extensively revised the manuscript. The new supplementary figures and tables substantiated the original statements. With all the additions and revisions, the authors have addressed all my concerns satisfactorily. This reviewer only has one more additional question. The authors mentioned that the thickness of the ferrofluid layer is “roughly tens of micrometers”. How did the authors estimate/measure the thickness? Is this value the thickness of the overlayer only? Or does it also include the thickness of the porous substrate?

Response: Thank you very much for the positive comment! The thickness (d) of the ferrofluid refers to the thickness of the overlayer. It was controlled by controlling the mass (m) of the overlayer ferrofluid. The depth of the surface structure of the substrate was in the nanoscale (Figure S2). Therefore, the ferrofluid infused in the structure was neglected. Since the area (A) of the substrate was known, we estimated the thickness by $d=m/\rho A$, where ρ is the density of the ferrofluid.

This information has been included in the Methods section.